# Evaluation of a Non-Parenteral Opioid Analgesia Protocol for Acute Sickle Cell Pain Episodes in Children

**DOI:** 10.3390/jcm8101728

**Published:** 2019-10-18

**Authors:** Paul Telfer, Filipa Barroso, Kim Newell, Jo Challands, Banu Kaya

**Affiliations:** 1Centre for Genomics and Child Health, Blizard Institute, Queen Mary University of London, London E1 2AT, UK; 2Department of Paediatric Haematology, Royal London Hospital, Bart’s Health NHS Trust, London E1 2ES, UK; filipabarroso@nhs.net (F.B.); kim.newell@nhs.net (K.N.); banukaya@nhs.net (B.K.); 3Department of Paediatric Anaesthetics, Royal London Hospital, Bart’s Health NHS Trust, London E1 1BB, UK; j.challands@nhs.net

**Keywords:** sickle, children, pain, opioid, oral

## Abstract

We evaluated a protocol comprising intranasal diamorphine (IND) combined with oral short and modified-release morphine for children at the emergency department (ED) with acute painful episodes of sickle cell disease (SCD). In a retrospective audit of 83 episodes in 38 children, the mean time between arrival in the treatment area and the administration of IND was 10 min (range <5 min to 1.39 h). IND was administered in <5 min in 43 (51.6%), and in <20 min in 75 (90.4%) episodes. Persisting pain, requiring background analgesia with modified-release oral morphine, was required in 25 (30.1%) episodes. Inadequate control of pain requiring a switch to intravenous morphine PCA was required in eight episodes in four patients. Acute chest syndrome (ACS) developed in four of 83 episodes (4.8%, 95% CI 0.2–9.4%) and in four of 38 children (10.5%, 95% CI 0.7–20.5%). In conclusion, this protocol enabled the rapid administration of strong opioid analgesia in an ED setting, and may reduce the short and long-term adverse effects associated with parenteral opioids in children. There was no evidence of an increased incidence of ACS associated with use of oral morphine.

## 1. Introduction

The painful crisis is the most frequent acute complication of sickle cell disease (SCD) in children and adults [1]. Severe episodes often require management in hospital with strong opioid analgesics (morphine and opioid drugs of similar or increased potency, such as diamorphine, oxycodone, or hydromorphone), and close monitoring for adverse effects, particularly over-sedation and respiratory depression. Delays in triaging, clinical assessment, and administering analgesia in emergency department (ED) result in sub-optimal pain management and might be associated with increasing pain levels with a prolonged duration of the episode and hospital admission [2,3,4,5]. In 2012, the National Institute of Health Care Excellence (NICE) in the UK published an evidence-based guideline in 2012, recommending that the first dose of strong opioid analgesia should be given within 30 min of presentation [6]. An National Heart, Lung and Blood Institute (NHLBI) guideline in the USA, published in 2014, recommended that the first dose should be given within 30 min of triage and within 60 min of the first presentation [7]. The time until first analgesia is regarded as a standard of care, but it has been repeatedly shown that timeliness is difficult to achieve [8]. 

One problem arising from current guidelines is the recommendation that the first opioid dose should be given parenterally [7]. There may be difficulties in venous access, and in preparation and double-checking of intravenous solutions. Intravenous cannulation and intramuscular or subcutaneous injections are all uncomfortable for children. 

We hypothesize that continued parenteral analgesia with bolus or continuous i.v. infusions may lead to a prolongation of, and dependence on, hospital based therapy for the management of acute painful episodes. This might be mitigated by use of oral opioids. In a post-hoc analysis of a randomized controlled trial conducted to evaluate the effect of i.v. magnesium on the duration of acute pain episode, the earlier initiation of oral opioids was associated with a shorter length of stay and improved health related quality of life at discharge. The authors of this study cite several articles in the surgical literature supporting these findings [3]. Furthermore, a small RCT comparing continuous i.v. infusion with oral controlled release morphine for continued opioid analgesia during the pain episode in children showed a reduction in the length of hospital stay with oral opioids, although this difference was not statistically significant [9]. These data support the use of oral analgesia regimes as an alternative to parenteral regimes, but it is clear that further studies are needed and should start with preliminary work to develop safe and effective protocols, which could then be used as comparators in future randomized controlled trials.

A post-hoc analysis of the study of Jacobson et al. suggested that the incidence of acute chest syndrome (ACS) was higher in the oral morphine group, and it was suggested that this might be due to increased levels of morphine and morphine 6-glucuronide, an active metabolite [10]. The dosing for oral modified-release morphine in this trial could be criticized for being too high (1.9 mg/kg bd) and being offered to all children, including those who are relatively opioid naïve, without an assessment of the need for background analgesia. 

Diamorphine is a potent opiate that is highly soluble, enabling its preparation at a high concentration. It is rapidly effective when given by the intranasal route in children with acute pain related to fractures, with the duration of analgesic effect from 20–30 min [11]. We previously reported a pilot study on intranasal diamorphine (IND) for acute pain management in SCD, showing rapid, effective analgesia [12]. We subsequently developed an analgesia protocol for children regimes using IND for immediate analgesia, supplemented by oral morphine for sustained pain relief until crisis resolution. Modified-release morphine was only prescribed after an assessment of the need for background analgesia at six-hours. When required, it was administered at an initial dose of 1 mg/kg bd, nearly 50% lower than in the Jacobson study. The full protocol is shown in Figure 1. In this study, we report an evaluation of the safety, efficacy, and tolerability of the protocol based on an in-house audit after approximately two years of implementation. 

## 2. Methods

Children with severe pain, as assessed using the College of Emergency Medicine Best Practice Guidelines for the assessment of pain in children [13], and whose pain had not responded to routine analgesia administered at home (paracetamol, ibuprofen, and codeine or dihydrocodeine), were administered IND and oral morphine, and were subsequently managed according to the protocol depicted in Figure 1. 

We tried to ensure rapid triage, assessment, and administration of the first analgesic dose by educating parents, receptionists, and medical and nursing staff. The receptionist was encouraged to ‘fast-track’ children with SCD through to the ‘majors’ treatment area (equivalent to US level 1 or 2 [2]). Nursing staff ensured rapid weighing and assessment of vital signs. Medical staff were generally available to make a brief medical assessment, and promptly prescribed the first dose of IND and oral morphine, which was then prepared and administered rapidly by ED nursing staff. Evaluation of the pain score and adverse effects were done at t = 0, 15, 30, and 60 min, hourly thereafter for the first six-hours and then four-hourly. The British Association of Emergency Medicine pediatric pain assessment tool was used [13]. 

We retrospectively audited consecutive admissions over a 12-month period using a standard form to collect data from clinical records. Patients aged 3–16 with SCD (HbSS, HbSC, or HbS/ beta thalassaemia) were included. Exclusion criteria were respiratory depression, shock, and airway compromise. The time until first analgesia was recorded as the time difference between arriving in the treatment area and receiving IND. The time until pain relief was defined as mild or no pain for at least six-hours. Patient acceptability was assessed prospectively in a subset of children or parents with a questionnaire devised within our service for the purpose of the audit. The questionnaire had not been formally validated. The evaluation of the full analgesia protocol and patient questionnaire were considered audit projects, and were approved by our institutional audit office.

## 3. Results

There were 96 presentations with acute pain, and IND was given in 83 episodes in 38 children. In the remaining 13 (13.5%), IND was not given, either because the pain was mild and could be managed with non-opioid analgesics, or parents requested oral morphine only. For those treated with IND, the median age (IQ range) at presentation was 12 years (10–15) and 23 (60.5%) were male. Thirty-six had HbSS and 2 HbSC. Eighteen children had one admission, seven had two, six had three, and seven had four or more.

The mean time interval between arrival in the treatment area and the administration of IND was 10 min (Median, <5 min, IQ range <5 to 8.3). IND was administered before five-minutes in 43 (51.6%), and before 20 min in 75 (90.4%) episodes. In five episodes, delay to IND administration was more than 60 min. Two of these episodes were complicated by severe anaemia that required urgent transfusion. For the remaining three episodes, the cause of delay in administering IND was not clear.

The median total dose of oral morphine was 1.6 mg/kg (IQ range 0.8 to 4.3), equivalent to four doses. Controlled-release oral morphine was required in 25 (30.1%) episodes. For these, the median total dose was 6.1 mg/kg (IQ range 4.8 to 8.2). For eight (9.6%) episodes in four patients, i.v. morphine via patient controlled analgesia pump (PCA) was required. In 72 evaluable episodes, the median time to pain relief was 20 h (IQ range 12 to 70). For all admissions, the median duration of admission was three-days (IQ range 2 to 6). 

Side effects were mostly mild and manageable by dosage reduction. 17.4% had constipation, 8% vomiting, 15% pruritis, and 1% excessive drowsiness. There were no life-threatening events attributable to the study medication. ACS was seen in four of 83 episodes (4.8%, 95% CI 0.2–9.4%) and in four of 38 children (10.5%, 95% CI 0.7–20.5%) and required either simple or exchange transfusion. 

The questionnaire was filled in by 21 subjects (17 children, 4 parents). The results are shown in Table 1. In general, the majority of children were satisfied with IND in regard to the speed of analgesia, and the route of administration. Additionally, a high percentage were satisfied with the management of pain as a whole, but felt that their hospital admission was too long.

## 4. Discussion

This study shows that IND can be safely combined with oral morphine to provide safe analgesia without the need for intravenous opiates. There was no evidence of an increased risk of ACS. The time to first analgesia was well within the standards of the NICE and NHLBI guidelines [6,7], and compares very favorably with the time to analgesia reported in studies of adult and paediatric ED departments in the USA [2,4,5,14]. Delays in obtaining intravenous access were avoided, and ED nursing staff could be trained to prepare and administer IND very rapidly. The median duration of admission (three days) was also less than in other reported studies in children, where figures of around five-days have been reported [3,5], suggesting that an effective oral analgesia protocol, when correctly implemented, provides the opportunity for an earlier discharge so that residual pain can be managed at home without the need to ‘wind down’ intravenous analgesia in hospital. There was an apparent discrepancy noted between the median time to pain relief and the length of stay. This was explained by the recurrence of pain after initially reaching the definition of pain relief. Fluctuations in pain were well described during the course of acute pain episodes. In some cases, pain relief was initially achieved, but recurred after withdrawal from analgesia. 

ACS was seen in four of 83 episodes (4.8%, 95% CI 0.2–9.4%) and in four of 38 children (10.5%, 95% CI 0.7–20.5%). Unfortunately, we do not have retrospective data to evaluate the rate of ACS in children managed at our hospital prior to the study. This was because the previous analgesia intravenous morphine protocol was gradually phased out and replaced by the current protocol, while the audit was undertaken about 12 months after full implementation. Rates of ACS in previous studies have been variable, ranging from 3% to 57%. Part of the variability may be due to differences in criteria for the definition of ACS. More recent and larger RCT’s of treatment interventions to attenuate vaso occlusive crises have provided more consistent figures of between 3% and 18% [15,16,17,18], and our results would be consistent with these rates. A significantly higher rate of ACS was observed with oral compared to i.v. morphine, and was reported in a post-hoc analysis of the study by Jacobson et al. [10]. They reported a rate of 12 out of 21 episodes treated in the oral morphine arm (57.1%, 95% CI 35.9–72.3%), compared to four out of 23 in the intravenous morphine arm (17.4%, 95% CI 1.9–32.9%). The pharmacokinetic data in this study led the authors to conclude that the risk of acute chest syndrome was significantly associated with high systemic exposure to morphine and its active metabolite morphine-6-glucuronide after oral administration of controlled-release morphine. The dose of controlled-release morphine in their study (1.9 mg/kg twice daily) was excessive, and nearly two times the dose in our protocol (1 mg/kg twice daily). Our results suggest that, at the lower dose used in our protocol, oral morphine does not increase the risk of ACS and we would disagree with the statement of the authors that, because of the high risk of ACS, oral sustained release morphine is not safe for treatment of acute pain in children with SCD. 

In our study, only 30% of episodes required controlled release oral morphine for background analgesia, and 10% of cases required a switch to intravenous morphine for very severe episodes of pain. We recommend that the option to switch to intravenous morphine is available as part of this analgesia protocol. 

In conclusion, combined IND and oral morphine could be implemented more widely in countries where diamorphine is licensed for analgesic use. Alternative opiates and opiate sparing agents also need to be evaluated to optimize the management of acute sickle cell pain in children. 

## Figures and Tables

**Figure 1 jcm-08-01728-f001:**
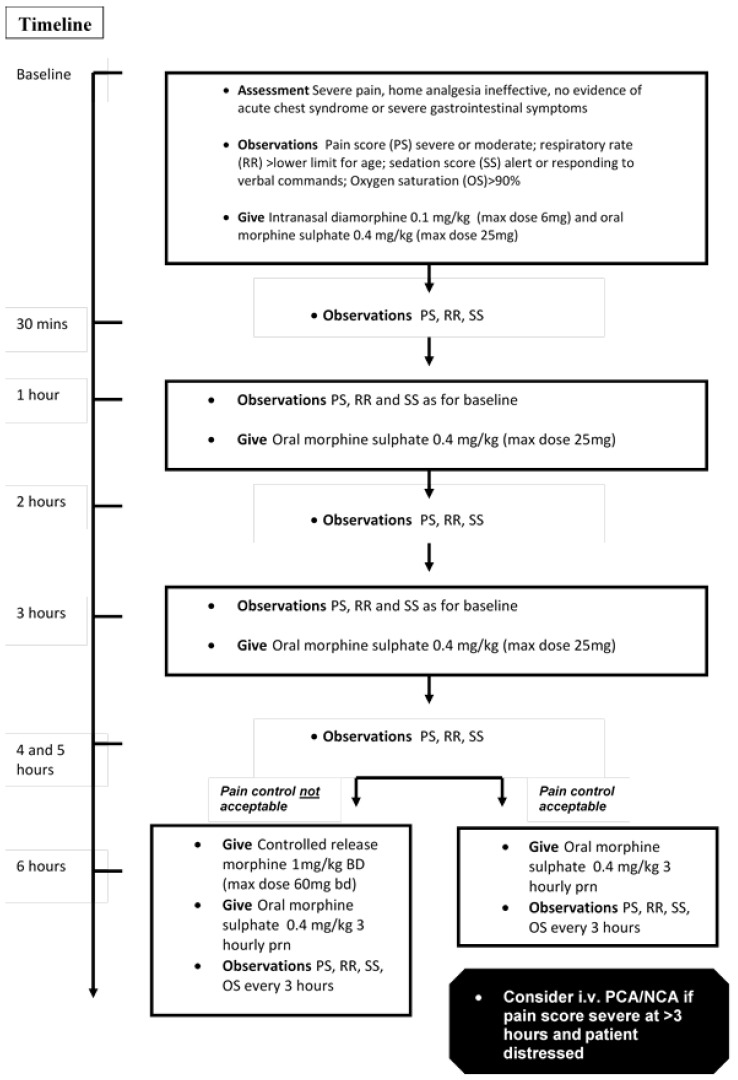
Full analgesia protocol.

**Table 1 jcm-08-01728-t001:** Patient/parent questionnaire results (figures are percentages).

**Intranasal Diamorphine**				
1. How quickly were you given pain-killers after you arrived in casualty	Immediate	5 min	10–20 min	>20 min
	38	24	33	5
2. How satisfied are you with the amount of time you had to wait to be given painkillers?	Satisfied	Sat/Disat	Disatisfied	
	71	19	10	
3. How long did it take for the painkiller in the nose to work to reduce the pain?	Immediate	5 min	10 min	20 min
	6	46	24	24
4. How satisfied are you with the amount of time it took for the pain killer to reduce your pain	Satisfied	Sat/Disat	Disat	
	74	26	0	
5. I find the painkiller in the nose	Comfortable	Uncomfortable		
	65	35		
**Pain Management**				
1. How well do you think people assess your level of pain when you are on the ward?	More pain than I feel	Correct	Less pain than I feel
	18	59	23	
2. What do you think about the painkillers given during your hospital admission?	Too strong	Correct	Too weak	
	5	76	19	
3. Did the nurses give you painkillers when you needed them?	Always	Usually	Sometimes	Never
	48	38	14	0
4. How satisfied are you with the way your pain was treated during your hospital admission?	Satis	Satis/Dist	Disat	
	81	14	5	
5. Compared to what I expected, the length of my hospital admission was	Longer	As I expected	Shorter	
	60	25	15

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
