# Peer review of "Evaluation of a Non-Parenteral Opioid Analgesia Protocol for Acute Sickle Cell Pain Episodes in Children"

_jcm, 2019, doi:10.3390/jcm8101728_

Round 1
Reviewer 1 Report
The manuscript presents a retrospective review of a protocol in which children with sickle cell disease and pain crisis presenting to the emergency department were given an intranasal opioid combined with oral morphine upon arrival. Time to first dose of pain medication is an important issue in the treatment of pain crisis in people with sickle cell disease, and while IV pain medication can result in achieving pain control quickly, administering it in a timely manner can be challenging. This study showed that intranasal opioids can be administered rapidly too children and can lead to adequate analgesia. This study thus addressed an important topic and would be of significant value to clinicians who care for children with sickle cell disease.
The authors state that “Continued parenteral analgesia with bolus or continuous iv infusions may also lead to a prolongation and dependence on hospital based therapy during childhood and adolescence with potential complications later in adulthood.” However, they have no evidence or references for this statement. This is a contentious statement in the light of the current opioid crisis and the efforts of many health care leaders in sickle cell disease to have sickle cell disease made an exception to many of the new rules restricting opioid use. Such a statement should only be made if it can supported by evidence in the current literature. It is unclear what the starting time is for the time to administration of IND. The statement “on arrival in causality” is not defined. It is unlikely that patients were given IND “immediately” as they had to be registered and triaged and seen by a health care professional who would then place the medication order. What was the average time for this to take place? If pain was relieved in a median of 20 hours why was average length of admission 3 days? The authors state there was no increased risk of acute chest, however no statistical results are presented. There is also no comparison of acute chest rates in the hospital previous to this study. The study only evaluated 33 children (though multiple episodes per child). Was the study powered to evaluate for acute chest rate? If the 95% confidence interval of rate of acute chest per this study were stated this would be helpful. If the study is underpowered the statement that oral morphine does not increase rate of acute chest should be less definitive.
Minor Comments:
ACD is not defined in the abstract Strong opioid is not clearly defined, what is a strong vs a not strong opioid? Font size varies throughout the manuscript Statistical methods should be discussed in results, even if no comparisons were made decisions were made to report percentiles in some cases and mean in others. These decisions should be explained. MST is not defined. Top up should be defined, is this a simple transfusion?
Reviewer 2 Report
This was a nice report on a protocol for use of intranasal diamorphone for children with a vaso-occlusive crisis. The authors found rapid administration time and no significant adverse events. I do think that it is a stretch to say that this study proves that IND may reduce short and long term adverse effects associated with parenteral opiates. This study was NOT powered to even look at adverse events. I think the most you can say is that there was no adverse effects in this study. This study also does not even mention long term effects
Background:
Line 47 (and 90) - IV is not capitalized.
Line 51 - saying a non-significantt reduction seems sneaky. IT is implying there was a reduction to some degree; however, given our statistical norms no significant reduction = NO REDUCTION.
Methods: I think there needs to be more explanation to hear explain the general procedure to make this more applicable for a broad audience. For instance - in America, we never admit patients with a VOC unless they need IV pain meds. It took me a while to recognize that these patients were getting admitted for oral morphine. at one point are people hospitalized? At what pain score do you give a dose of oral morphine? There needs to be more clear elucidation of what is the protocol that was used
line 79 - what is meant by " the final category"
Line 80 - 'a' not 'an'
Results
Line 89 - what is MST?
Table - can you state that these are percentages
Results -
How can you say there is no increased risk of ACS? You don't say what baseline risk of ACS is... provide citation for this
